# Low-Cost Assessment Method for Existing Adjacent Beam Bridges

Siqi Wang [1], Jinsheng Du [1,*] and Han Su [2]

1   School of Civil Engineering, Beijing Jiaotong University, Beijing 100044, China
2   Department of Civil Engineering, Tsinghua University, Beijing 100084, China
*   Correspondence: jshdu@bjtu.edu.cn; Tel.: +86-10-51688247

**Abstract:** Damage in grouted joints is an unavoidable early disease in adjacent box beam bridges and hollow-core slab bridges. Joint damage will lead to degradation of the transverse load transmission capacity of the bridge, causing beams of the bridge superstructure to bear loads higher than the designed value, and eventually fail prematurely. Precise assessment of bearing–capacity degradation degree of adjacent box beam bridges and hollow-core slab bridges that are of great number is the keypoint to maintaining the serviceability of traffic network. The current specifications regard grouted joints as individual components and cannot correctly assess the degradation degree of bearing capacity caused by joint damage. In this paper, the traditional hinge connected beam method is improved by modifying deformation compatibility conditions at grouted joints. By using a modified hinge connected beam method, the relationship of joints at different locations with the lateral load distribution factor (LLDF) is analyzed. Based on analysis results, this paper proposes a new low-cost assessment method and a new assessment index that can utilize visual inspection results. Based on the concept of standard deviation, the proposed method assesses the degradation degree of the lateral load transmission performance of bridge superstructures by calculating the variation in LLDFs of beams, which is expressed by the lateral load distribution performance rating number $LDN$. The proposed method is applied to three real bridges. The accuracy of the calculation results is verified by comparing the ranking of $LDNs$ of three bridges with the ranking of the variation degrees of lateral deflection influence lines of three bridges obtained from static-load test results.

**Keywords:** adjacent box beam bridges; hollow-core slab bridges; lateral load distribution factors; bridge structural assessment





## 1. Introduction

Adjacent precast concrete girder bridge systems are a common form for short-to-medium-span highway bridges, with adjacent box beam bridges (ABBB) and adjacent hollow-core slab bridges (AVSB) being two prevalent types [1]. Such bridges can be described as structures composed of precast box beams or hollow-core slabs that abutting one another with keyways reserved, linked laterally by longitudinal shear keys filled with grouting, and overlaid with concrete or asphalt pavement.

The design procedure of transverse connections varies from country to country. As shown in Figure 1 [2], in some cases of the United States, Japan, and South Korea, end and intermediate diaphragms with transverse ties that consist of post-tensioning strands/tie-bars are installed in the box beams and hollow-core slabs to strengthen transverse stiffness and improve the load transfer mechanism [3,4]. In China, instead of diaphragms and transverse post-tensioning, adjacent beams are connected by overlapping loop reinforcements that are extended from girders into shear keys and arranged closely along the longitudinal direction of the bridge [2].

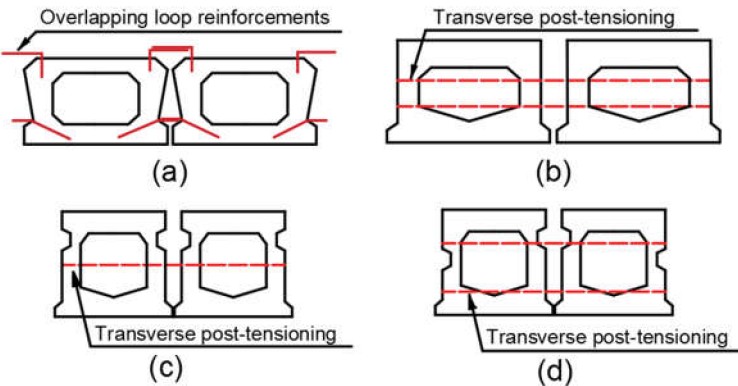

**Figure 1.** Comparison of adjacent beams with different transverse connections [2]: (**a**) China, (**b**) Japan, (**c**) the United States, (**d**) South Korea.

Therefore, in order to avoid confusion and for the convenience of expression, adjacent box beam bridges (ABBB) and adjacent void slab bridges (AVSB) will be represented by adjacent rectangular hollow-beam (ARHB) bridges in this paper.

The condition of transverse connections greatly affects the performance of in-service precast ARHB bridges [5]. However, a recurring issue occurring in this type of bridge is cracking in the grouted joints (shear keys and interface between shear keys and beams) between adjacent beams, which will result in reflective cracks forming in pavements and leakage of chloride-laden water, which will penetrate into shear keys and the girders, causing corrosion of the reinforcement (prestressed and non-prestressed) [4]. These will cause the loss of load transfer in the lateral direction, eventually leading individual girders to endure live loads larger than the design live loads and then fail.

Timely maintenance and repair are the keys to ensuring the in-service performance of ARHB bridges and preventing collapse accidents, while the maintenance and repair arrangements are based on bridge performance assessment results.

The procedure of bridge performance assessment is to gain the bridge damage information first, and then evaluate the bridge performance by corresponding measurement indicators, which can be categorized into two groups: one is index values calculated from a defined formula using input data, and the other is whether bridges meet defined criteria or not [6].

At present, the acquisition methods of bridge-damage information include routine visual inspection, static and/or dynamic load testing methods, finite-element (FE)-model-based methods, and data-driven methods. Visual inspection has become the default methodology of routine inspection [7] due to its advantages of economic, convenient operation, and no interference with traffic [8]. In addition, bridge condition assessment results based on visual inspections also serve as the basis for determining whether more advanced and detailed inspections are required.

Data-driven methods, including statistical analysis or pattern recognition, machine learning, and deep learning methods, have attracted the attention of scholars and obtained some achievements [9–11], while no research results on grouted joints of adjacent beam bridges have been found.

As more mature technologies than data-driven methods, static and/or dynamic load testing methods and finite-element (FE)-model-based methods can provide more accurate and objective information than visual inspection on damage detection, quantification, and localization, so it is the research field most concerned by scholars. Zhan et al. proposed a damage-identification method based on vehicle–bridge interaction analysis and the model-updating method; the existence and severity of damage in adjacent box beam bridges can be obtained by solving the minimization problem of the objective function that is constructed using the coherence function of the response spectrum (CFRS) index and frequency residual vectors [2]. Xu et al. used structural vibration information collected by accelerometers and

dynamic strain gauges installed at the bottom of girders in mid-span to identify transverse mode shape of the multi-beam system to evaluate the transmission performance of grouted joints of assembled concrete multi-girder bridges [12]. Hu et al. proposed a hybrid method using physical models and vision-based measurements, and demonstrated the applicability of relative displacement ratio as the damage index of hollow-core slab bridge grouted joints by establishing a simplified spring–mass system wherein the relative displacement ratio of inspected hinge joints was extracted by computer vision-based multi-point displacement measurement methods [13]. Dan defined the lateral collaborative performance indicator of assembled beam bridge grouted joints as the ratio of the linear correlation coefficients of longitudinal strain of two adjacent beams; the effectiveness and accuracy of the indicator have been proved by the good agreement of strain data analysis results and actual bridge observations [14]. Based on the literature review, it can be seen that these methods have disadvantages, such as that the static load testing method will cause traffic blocking, dynamic load testing methods are susceptible to noise interference, and finite-element (FE)-model-based methods are restricted by the accuracy of the structure model. Moreover, the sensor instrumentation costs of load testing methods make them only applicable to the special inspection of specific bridges, rather than the routine inspection of the large number of small-to-medium-span bridges.

The performance assessment based on visual inspection regards grouted joints of adjacent rectangular hollow-beam (ARHB) bridges as individual elements. Condition indices of grouted joints are multiplied by component weight factors and added with other weighted-element condition indices to constitute the condition index of the whole bridge. For example, according to *AASHTO (2003)* [15], the health index H of the entire bridge is calculated by the weighted average of the element health indexes, which are the ratio of the summation of the products of the corresponding coefficient and the element quantities in each condition state to the total quantity of the element. Furthermore, the element number can be found in AASHTO CORE. In China, according to *Standards for Technical Condition Evaluation of Highway Bridges (JTG/T H21-2011)* [16], grouted joints as well as diaphragms are classified as general members of superstructure with the other two parts of the superstructure being load-carrying members and bearings, while according to the *Technical standard of maintenance for city bridge (CJJ99-2017)* [17], grouted joints are classified as a part of the transverse connections and evaluated by the penetration degree of longitudinal cracks on the bridge decks. In general, the condition score of the structure can be obtained by deducting the scores represented by the detected member damages. Although shear keys are not load-bearing members, the failure of shear keys will decimate the load distribution, causing individual girders to be exposed to live loads that are greater than the designed value. Performance assessment results based on visual inspection and current specifications cannot fully reflect the important impact of grouted joints on the bearing capacity of bridges. Since joint damage is common in ARHB bridges, in order to avoid premature degradation of the bridge due to bearing loads higher than the designed value [13], a low-cost assessment method that can be combined with visual inspection is required to evaluate the degradation of load transverse transmission performance caused by joint damage and to arrange maintenance.

When the transverse load transmission performance of the bridge deteriorates due to joint damage, the influence line of the lateral load distribution will become steep, and the lateral load distribution factor (LLDF) of the beam bearing the load will become larger. Therefore, the degradation degree of the ARHB bridge can be evaluated by variation in the LLDF due to joint damage. In China, the hinge connected beam method is the most commonly used design method for ARHB bridges to calculate lateral load distribution factors (LLDFs). By introducing the relative displacement of adjacent beams caused by joint damage, the modified method can be used for the calculation of existing ARHB bridges.

Therefore, the research in this paper proposed a new evaluating indicator, namely, the lateral load distribution performance rating number $LDN$, to evaluate the degradation degree of ARHB bridges due to joint damage. Section 2 presents the traditional hinge

connected beam method and the modified method that considers joint damage, and analyzes the influence of the damaged joint at different positions on the LLDF of the beam. Section 3 illustrates the applicability of the modified hinge connected beam method on ARHB bridges with diaphragms and transverse prestressed ties by analyzing the shear transmission mechanism at grouted joints. Section 4 introduces the new assessment method and evaluating indicator of the degradation degree of transverse load transmission performance based on joint damage proposed in this paper in detail. The proposed method and indicator are exemplified through a case study, and the degradation degree calculated by the proposed indicator is compared with the degradation degree calculated by static load test results. Conclusions are drawn in Section 6.

## 2. Influence of Damage Joints on LLDFs Based on the Modified Hinge Connected Beam Method

### 2.1. Hinge Connected Beam Method

Despite the fact that computer technology and numerical simulation methods have been widely used in bridge design and analysis, simplified analysis methods are still favored by engineers. That is because the simplified methods are time-efficient and can be used to verify the results of numerical simulation [18].

At present, many bridge design codes, such as the *AASHTO Load and Resistance Factor Design (LRFD) Bridge Design Specifications (9th edition, 2020)* [19], the *Canadian Highway Bridge Design Code (2019)* [20], and the *General Specifications for Design of Highway Bridges and Culverts (2015)* [21], adopt simplified methods. That is, they regard the lateral and longitudinal effects of live loads as uncoupled phenomena to transform the spatial stress problem into the plane stress problem by lateral load distribution factors [22,23]. The lateral load distribution factor LLDF is defined as the ratio of the live load carried by the individual beam and the applied live load, and expressed as the ratio of the load effect of the individual beam and the total load effects of all beams.

The hinge connected beam method is a frequently used simplified method for calculating LLDFs of adjacent rectangular hollow-beam (ARHB) bridges in China. This method is derived from the force method and is applicable to fabricated T-beam bridges without intermediate diaphragms, connected only by steel plates or reinforcements between flanges, or fabricated bridges that are connected only by cast-in-situ bridge decks, or adjacent rectangular hollow-beam (ARHB) bridges that are connected by cast-in-situ joints and without diaphragms and transverse prestressed ties.

The basic assumptions of the hinge connected beam method are as follows [24,25]:

(1) The adjacent rectangular hollow-beam (ARHB) bridge is simplified as several beams in parallel and horizontally hinged with each other. Each beam is a Euler–Bernoulli uniform beam and simply supported at the end;

(2) Only analyzing shear force transmission between adjacent beams, leaving the transfers of normal stress, bending moment, and longitudinal shear force out. Assuming that each beam has only two degrees of freedom that are vertical displacement and longitudinal rotation. Ignoring torsional effect, lateral deflection deformation, and section deformation;

(3) The adjacent beams have no relative displacement at joints, and the stiffness of joints is infinite.

As shown in Figure 2a–c, for an ARHB bridge with $n$ beams and $n-1$ joints, when applying the dimensionless external load $p = 1$ on the central axis of one of the beams, a pair of vertical shear forces with equal size and opposite direction will be generated in each joint.

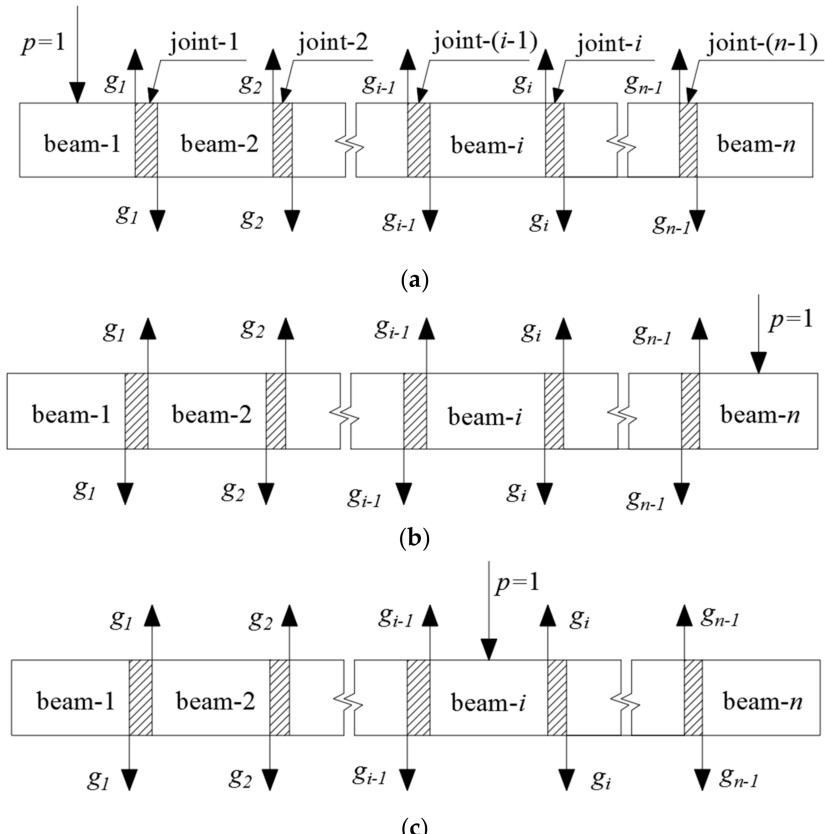

**Figure 2.** (**a**) Force analysis diagram when external load $p = 1$ applied on beam-1. (**b**) Force analysis diagram when external load $p = 1$ applied on beam-$n$. (**c**) Force analysis diagram when external load $p = 1$ applied on beam-$i$.

According to the principles of force balance and the deformation compatibility condition that the vertical relative displacement of two adjacent beams at joints is zero, the canonical equation can be obtained, as shown in Equation (1):

$$
\begin{cases}
\delta_{11}g_1 + \delta_{12}g_2 + \cdots + \delta_{1,n-1}g_{n-1} + \delta_{1p} = 0 \\
\delta_{21}g_1 + \delta_{22}g_2 + \cdots + \delta_{2,n-1}g_{n-1} + \delta_{2p} = 0 \\
\ \vdots \qquad\quad \vdots \qquad\quad \ddots \qquad\ \vdots \qquad\quad \vdots \qquad \vdots \\
\delta_{n-1,1}g_1 + \delta_{n-1,2}g_2 + \cdots + \delta_{n-1,n-1}g_{n-1} + \delta_{n-1,p} = 0
\end{cases}
\tag{1}
$$

where $g_j$ represents the vertical shear force at joint-$j$, $\delta_{ij}$ represents the vertical displacement at joint-$i$ caused by the vertical shear force $g_j$, and $\delta_{ip}$ represents the vertical displacement caused by external load $p$ at joint-$i$.

$\delta_{ij}$ is composed of the displacement at the center of the beam and the rotation angle generated by $g_j$. When the external load $p = 1$ is applied on the exterior beams, $\delta_{ij}$ is shown as Equation (2).

$$
\begin{cases}
\delta_{11} = \delta_{11} = \cdots = \delta_{n-1,n-1} = 2 \times \left(\omega + \frac{b}{2} \times \varphi\right) \\
\delta_{12} = \cdots = \delta_{n-2,n-1} = \delta_{21} = \cdots = \delta_{n-1,n-2} = -\left(\omega - \frac{b}{2} \times \varphi\right) \\
\delta_{ij} = 0 \quad when\ i < j - 1,\ i > j + 1
\end{cases}
\tag{2}
$$

where $\omega$ represents the displacement at the center of the beam generated by $g_j$ and expressed by Equation (3), and $\varphi$ represents the rotation angle generated by $g_j$ and expressed by Equation (4):

$$
\omega = \frac{pl^4}{\pi^4 EI}
\tag{3}
$$

$$\varphi = \frac{pbl^4}{2\pi^2 GI_T} \tag{4}$$

where $p$ represents external load, $b$ represents the beam width, $l$ represents the span length of simply supported beam, $E$ and $G$ represent elastic modulus and rigid modulus of the beam, respectively, and $I$ and $I_T$ represent the bending moment of inertia and the torsional moment of inertia of the beam, respectively.

When the external load $p = 1$ is applied on the interior beams, here beam-$k$, $2 \le k \le n - 1$, $\delta_{ij}$ is shown as Equation (5).

$$\begin{cases} \delta_{11} = \delta_{11} = \cdots = \delta_{n-1,n-1} = 2 \times \left(\omega + \frac{b}{2} \times \varphi\right); \\ \delta_{k-1,k} = \delta_{k,k-1} = \omega - \frac{b}{2} \times \varphi; \\ \delta_{12} = \cdots = \delta_{n-2,n-1} = \delta_{21} = \cdots = \delta_{n-1,n-2} = -\left(\omega - \frac{b}{2} \times \varphi\right) \\ \qquad \text{where excluding } \delta_{k-1,k} = \delta_{k,k-1}; \\ \delta_{ij} = 0 \qquad \text{when } i < j - 1,\ i > j + 1; \end{cases} \tag{5}$$

$\delta_{ip}$ of external load $p = 1$ applying on exterior beam-1 and beam-$n$ is shown as Equation (6a), and on exterior beam-$k$ is shown as Equation (6b).

$$\begin{cases} \delta_{ip} = -\omega \ i = 1 \ or \ n \\ \delta_{ip} = 0 \ the \ others \end{cases} \tag{6a}$$

$$\begin{cases} \delta_{ip} = -\omega \ i = k - 1 \\ \delta_{ip} = -\omega \ i = k \\ \delta_{ip} = 0 \ the \ others \end{cases} \tag{6b}$$

Substitute Equations (2)–(6a,b) into Equation (1), and divide both sides of Equation (1) by $\omega$ at the same time, and simplified forms of Equation (1) can be obtained and shown as Equation (7a–c).

When the external load $p = 1$ is applied on exterior beam-1, the simplified form of Equation (1) is shown as Equation (7a):

$$\begin{cases} 2(1 + \gamma)g_1 - (1 - \gamma)g_2 = 1 \\ -(1 - \gamma)g_1 + 2(1 + \gamma)g_2 - (1 - \gamma)g_3 = 0 \\ \vdots \quad \vdots \quad \ddots \quad \vdots \quad \vdots \quad \vdots \\ -(1 - \gamma)g_{n-3} + 2(1 + \gamma)g_{n-2} - (1 - \gamma)g_{n-1} = 0 \\ -(1 - \gamma)g_{n-2} + 2(1 + \gamma)g_{n-1} = 0 \end{cases} \tag{7a}$$

When the external load $p = 1$ is applied on exterior beam-$n$, the simplified form of Equation (1) is shown as Equation (7b):

$$\begin{cases} 2(1 + \gamma)g_1 - (1 - \gamma)g_2 = 0 \\ -(1 - \gamma)g_1 + 2(1 + \gamma)g_2 - (1 - \gamma)g_3 = 0 \\ \vdots \quad \vdots \quad \ddots \quad \vdots \quad \vdots \quad \vdots \\ -(1 - \gamma)g_{n-3} + 2(1 + \gamma)g_{n-2} - (1 - \gamma)g_{n-1} = 0 \\ -(1 - \gamma)g_{n-2} + 2(1 + \gamma)g_{n-1} = 1 \end{cases} \tag{7b}$$

When the external load $p = 1$ is applied on interior beam-$k$, where $2 \leq k \leq n - 1$, the simplified form of Equation (1) is shown as Equation (7c):

$$
\begin{cases}
2(1 + \gamma)g_1 - (1 - \gamma)g_2 = 0 \\
-(1 - \gamma)g_1 + 2(1 + \gamma)g_2 - (1 - \gamma)g_3 = 0 \\
\quad \vdots \quad\quad \vdots \quad\quad \ddots \quad\quad\quad \vdots \quad\quad\quad \vdots \quad\quad \vdots \\
-(1 - \gamma)g_{k-2} + 2(1 + \gamma)g_{k-1} + (1 - \gamma)g_k = 1 \\
(1 - \gamma)g_{k-1} + 2(1 + \gamma)g_k - (1 - \gamma)g_{k+1} = 1 \\
\quad \vdots \quad\quad \vdots \quad\quad \ddots \quad\quad\quad \vdots \quad\quad\quad \vdots \quad\quad \vdots \\
-(1 - \gamma)g_{n-3} + 2(1 + \gamma)g_{n-2} - (1 - \gamma)g_{n-1} = 0 \\
-(1 - \gamma)g_{n-2} + 2(1 + \gamma)g_{n-1} = 0
\end{cases}
\tag{7c}
$$

where $\gamma$ is bending and torsional parameter and can be expressed by Equation (8) for reinforced concrete ARHB bridges:

$$
\gamma = \frac{\varphi \frac{b}{2}}{\omega} \approx 5.8 \times \frac{I}{I_T} \times \left(\frac{b}{l}\right)^2
\tag{8}
$$

After obtaining $g_j$ by solving Equation (7c), the lateral load distribution factor $\eta_{ij}$ of each beam can be obtained by Equation (9a–c). $i$ of $\eta_{ij}$ is the number of the beam for which LLDF is being calculated, and $j$ of $\eta_{ij}$ is the number of the beam loaded by external load $p = 1$.

$$
\begin{cases}
\eta_{11} = 1 - g_1 \\
\eta_{i1} = g_{i-1} - g_i \quad \text{when } j = 1 \\
\eta_{n1} = g_{n-1}
\end{cases}
\tag{9a}
$$

$$
\begin{cases}
\eta_{1k} = g_1 \\
\eta_{ik} = |g_{i-1} - g_i| \\
\eta_{kk} = 1 - g_{k-1} - g_k \quad \text{when } 2 \leq j = k \leq n - 1 \\
\eta_{nk} = g_{n-1}
\end{cases}
\tag{9b}
$$

$$
\begin{cases}
\eta_{1n} = g_1 \\
\eta_{in} = g_i - g_{i-1} \quad \text{when } j = n \\
\eta_{nn} = 1 - g_{n-1}
\end{cases}
\tag{9c}
$$

The accuracy of the hinge connected beam method in calculating the LLDFs of ARHB bridges has been verified by a large number of real bridge experiments and monitoring data in China.

### 2.2. Modified Hinge Connected Beam Method Considering Joint Damage

The hinge connected beam method shown in Section 2.1 can only be used to calculate LLDFs of intact bridges with undamaged joints. This method assumes that there is no vertical relative displacement between adjacent beams at joints, and the stiffness of joints can be regarded as infinite. However, when damage exists in joints, adjacent beams will have a vertical relative displacement $\Delta$ at the damaged joint. The right side of Equation (1) established according to the deformation compatibility condition at the joint will no longer be zero. The canonical equations considering the relative displacement $\Delta_i$ caused by joint damages are shown in Equation (10):

$$
\begin{cases}
\delta_{11}g_1 + \delta_{12}g_2 + \cdots + \delta_{1,n-1}g_{n-1} + \delta_{1p} = \Delta_1 \\
\quad \vdots \quad\quad \vdots \quad\quad \ddots \quad\quad\quad \vdots \quad\quad\quad \vdots \quad\quad \vdots \\
\delta_{i1}g_1 + \delta_{i2}g_2 + \cdots + \delta_{i,n-1}g_{n-1} + \delta_{ip} = \Delta_i \\
\quad \vdots \quad\quad \vdots \quad\quad \ddots \quad\quad\quad \vdots \quad\quad\quad \vdots \quad\quad \vdots \\
\delta_{n-1,1}g_1 + \delta_{n-1,2}g_2 + \cdots + \delta_{n-1,n-1}g_{n-1} + \delta_{n-1,p} = \Delta_{n-1}
\end{cases}
\tag{10}
$$

where $\Delta_i$ represents the relative displacement at joint-$i$, $\Delta_i = 0$ when joint-$i$ is intact and $\Delta_i \neq 0$ when joint-$i$ is damaged.

The simplified form of Equation (10) is shown as Equation (11a–c) for external load $p = 1$ applied on exterior beam-1, exterior beam-$n$, and interior beam-$k$, respectively:

$$
\begin{cases}
2(1+\gamma)g_1 - (1-\gamma)g_2 = 1 + \frac{\Delta_1}{\omega} \\
-(1-\gamma)g_1 + 2(1+\gamma)g_2 - (1-\gamma)g_3 = \frac{\Delta_2}{\omega} \\
\vdots \quad \vdots \quad \ddots \quad \vdots \quad \vdots \quad \vdots \\
-(1-\gamma)g_{n-3} + 2(1+\gamma)g_{n-2} - (1-\gamma)g_{n-1} = \frac{\Delta_{n-2}}{\omega} \\
-(1-\gamma)g_{n-2} + 2(1+\gamma)g_{n-1} = \frac{\Delta_{n-1}}{\omega}
\end{cases}
\tag{11a}
$$

$$
\begin{cases}
2(1+\gamma)g_1 - (1-\gamma)g_2 = \frac{\Delta_1}{\omega} \\
-(1-\gamma)g_1 + 2(1+\gamma)g_2 - (1-\gamma)g_3 = \frac{\Delta_2}{\omega} \\
\vdots \quad \vdots \quad \ddots \quad \vdots \quad \vdots \quad \vdots \\
-(1-\gamma)g_{n-3} + 2(1+\gamma)g_{n-2} - (1-\gamma)g_{n-1} = \frac{\Delta_{n-2}}{\omega} \\
-(1-\gamma)g_{n-2} + 2(1+\gamma)g_{n-1} = 1 + \frac{\Delta_{n-1}}{\omega}
\end{cases}
\tag{11b}
$$

$$
\begin{cases}
2(1+\gamma)g_1 - (1-\gamma)g_2 = \frac{\Delta_1}{\omega} \\
-(1-\gamma)g_1 + 2(1+\gamma)g_2 - (1-\gamma)g_3 = \frac{\Delta_2}{\omega} \\
\vdots \quad \vdots \quad \ddots \quad \vdots \quad \vdots \quad \vdots \\
-(1-\gamma)g_{k-2} + 2(1+\gamma)g_{k-1} + (1-\gamma)g_k = 1 + \frac{\Delta_{k-1}}{\omega} \\
(1-\gamma)g_{k-1} + 2(1+\gamma)g_k - (1-\gamma)g_{k+1} = 1 + \frac{\Delta_k}{\omega} \\
\vdots \quad \vdots \quad \ddots \quad \vdots \quad \vdots \quad \vdots \\
-(1-\gamma)g_{n-3} + 2(1+\gamma)g_{n-2} - (1-\gamma)g_{n-1} = \frac{\Delta_{n-2}}{\omega} \\
-(1-\gamma)g_{n-2} + 2(1+\gamma)g_{n-1} = \frac{\Delta_{n-1}}{\omega}
\end{cases}
\tag{11c}
$$

Many scholars and specialists have proposed various models of the relative displacement $\Delta_i$. Li [26] utilized Equation (12) to express the relative displacement $\Delta_i$. As shown in Equation (12), $g_i$ represents the shear force at joint-$i$ generated by external load $p$, $\omega$ represents the maximum displacement of adjacent beams at the completely damaged joint when the dimensionless shear force $g_i = 1$, and $a\epsilon[0,1)$ represents the damage rate of joint-$i$.

$$
\Delta_i = -g_i a_i \omega
\tag{12}
$$

Zhou [27] used a spring connection model to simulate the shear force transmission in grouted joints of ARHB bridges and expressed the joint damage degree by reducing the joint rigidity. The author considered that the relative displacement $\Delta_i$ is inversely proportional to the rigidity of the shear keys at joint-$i$, and proposed a linear elastic model, which is shown as Equation (13):

$$
\Delta_i = -\frac{g_i}{k_i}
\tag{13}
$$

where $g_i$ represents the shear force at the joint-$i$, and $k_i$ represents the stiffness of joint-$i$.

However, the materials of damaged grouted joints are not linear elastic. The stress state is complicated and the force transmission mechanism is coupled by friction, mechanical interlock, and dowel action. A linear elastic model cannot accurately simulate the relationship between the shear force $g_i$ and relative displacement $\Delta_i$ of adjacent beams at joints. Despite this, it can still be determined that the joint damage degree is proportional to the relative displacement of adjacent beams at the joint. The greater the joint damage degree, the greater the relative displacement. Therefore, although it is impossible to obtain accurate relative displacement data to calculate LLDFs considering joint damage, it is feasible to utilize the relative displacement to analyze the influence of joint damage degree on the LLDFs.

Let $d_i = \left| \frac{\Delta_i}{\omega} \right|$ represent the damage degree of grouted joints. Equation (11a) can be written as Equation (14a). Similarly, Equation (14b,c) can be obtained.

$$\begin{cases} 2(1+\gamma)g_1 - (1-\gamma)g_2 = 1 - d_1 \\ -(1-\gamma)g_1 + 2(1+\gamma)g_2 - (1-\gamma)g_3 = -d_2 \\ \quad \vdots \quad \vdots \quad \ddots \quad \vdots \quad \vdots \quad \vdots \\ -(1-\gamma)g_{n-3} + 2(1+\gamma)g_{n-2} - (1-\gamma)g_{n-1} = d_{n-2} \\ -(1-\gamma)g_{n-2} + 2(1+\gamma)g_{n-1} = -d_{n-1} \end{cases} \tag{14a}$$

$$\begin{cases} 2(1+\gamma)g_1 - (1-\gamma)g_2 = d_1 \\ -(1-\gamma)g_1 + 2(1+\gamma)g_2 - (1-\gamma)g_3 = d_2 \\ \quad \vdots \quad \vdots \quad \ddots \quad \vdots \quad \vdots \quad \vdots \\ -(1-\gamma)g_{n-3} + 2(1+\gamma)g_{n-2} - (1-\gamma)g_{n-1} = d_{n-2} \\ -(1-\gamma)g_{n-2} + 2(1+\gamma)g_{n-1} = 1 + d_{n-1} \end{cases} \tag{14b}$$

$$\begin{cases} 2(1+\gamma)g_1 - (1-\gamma)g_2 = d_1 \\ -(1-\gamma)g_1 + 2(1+\gamma)g_2 - (1-\gamma)g_3 = d_2 \\ \quad \vdots \quad \vdots \quad \ddots \quad \vdots \quad \vdots \quad \vdots \\ -(1-\gamma)g_{k-2} + 2(1+\gamma)g_{k-1} + (1-\gamma)g_k = 1 + d_{k-1} \\ (1-\gamma)g_{k-1} + 2(1+\gamma)g_k - (1-\gamma)g_{k+1} = 1 + d_k \\ \quad \vdots \quad \vdots \quad \ddots \quad \vdots \quad \vdots \quad \vdots \\ -(1-\gamma)g_{n-3} + 2(1+\gamma)g_{n-2} - (1-\gamma)g_{n-1} = d_{n-2} \\ -(1-\gamma)g_{n-2} + 2(1+\gamma)g_{n-1} = d_{n-1} \end{cases} \tag{14c}$$

### 2.3. Influence of Damaged Joints at Different Locations on the LLDF

By combining Equation (9a–c) and Equation (14a–c), LLDFs containing the damage degree of damaged joints can be obtained and shown as Equation (15):

$$\eta_{ij} = \eta_{ij,intact} + \lambda_1 d_1 + \cdots + \lambda_i d_n + \cdots + \lambda_n d_n \tag{15}$$

where $\eta_{ij}$ is the LLDF of beam-*j*, $d_n$ is the damage degree of joint-*i*, and $\lambda_i$ is the influence degree of joint-*i* on the LLDF of beam-*j*.

The influence of damaged joints at different locations on LLDFs can be analyzed by comparing $\lambda_i$.

A simple example is used here to illustrate the relationship between the influence degree of joint-*i* on beam-*j* and the relative position of joint-*i* with beam-*j*.

Let the number of beams of an ARHB bridge be *n* = 7, and the bending and torsional parameter be $\gamma = 0.1$. When the external load $p = 1$ loads onto the exterior beam (beam-1), the shear force $g_i$ at joint-1 to joint-6 and the LLDFs of beam-1 to beam-7 are shown as Equation (16a,b):

$$\begin{cases} g_1 = 0.577 - 0.577d_1 - 0.299d_2 - 0.155d_3 - 0.079d_4 - 0.039d_5 - 0.016d_6 \\ g_2 = 0.299 - 0.299d_1 - 0.732d_2 - 0.379d_3 - 0.194d_4 - 0.095d_5 - 0.039d_6 \\ g_3 = 0.155 - 0.155d_1 - 0.379d_2 - 0.771d_3 - 0.395d_4 - 0.194d_5 - 0.079d_6 \\ g_4 = 0.079 - 0.079d_1 - 0.194d_2 - 0.395d_3 - 0.771d_4 - 0.379d_5 - 0.155d_6 \\ g_5 = 0.039 - 0.039d_1 - 0.095d_2 - 0.194d_3 - 0.379d_4 - 0.732d_5 - 0.299d_6 \\ g_6 = 0.016 - 0.016d_1 - 0.039d_2 - 0.079d_3 - 0.155d_4 - 0.299d_5 - 0.577d_6 \end{cases} \tag{16a}$$

$$\begin{cases} \eta_{11} = 0.423 + 0.577d_1 + 0.299d_2 + 0.155d_3 + 0.079d_4 + 0.039d_5 + 0.016d_6 \\ \eta_{21} = 0.278 - 0.278d_1 + 0.433d_2 + 0.244d_3 + 0.115d_4 + 0.056d_5 + 0.023d_6 \\ \eta_{31} = 0.144 - 0.144d_1 - 0.353d_2 + 0.392d_3 + 0.201d_4 + 0.099d_5 + 0.04d_6 \\ \eta_{41} = 0.076 - 0.076d_1 - 0.185d_2 - 0.376d_3 + 0.376d_4 + 0.185d_5 + 0.065d_6 \\ \eta_{51} = 0.040 - 0.040d_1 - 0.099d_2 - 0.201d_3 - 0.392d_4 + 0.353d_5 + 0.144d_6 \\ \eta_{61} = 0.023 - 0.023d_1 - 0.056d_2 - 0.115d_3 - 0.224d_4 - 0.433d_5 + 0.278d_6 \\ \eta_{71} = 0.016 - 0.016d_1 - 0.039d_2 - 0.079d_3 - 0.155d_4 - 0.299d_5 - 0.577d_6 \end{cases} \quad (16b)$$

When the external load $p = 1$ loads onto the interior beam (from beam-2 to beam-4), the LLDF of the beam with external load applied on is shown as Equation (17):

$$\begin{cases} \eta_{22} = 0.290 + 0.278d_1 + 0.433d_2 + 0.224d_3 + 0.115d_4 + 0.056d_5 + 0.023d_6 \\ \eta_{33} = 0.255 + 0.144d_1 + 0.353d_2 + 0.392d_3 + 0.201d_4 + 0.099d_5 + 0.040d_6 \\ \eta_{44} = 0.248 + 0.076d_1 + 0.185d_2 + 0.376d_3 + 0.376d_4 + 0.185d_5 + 0.076d_6 \\ \eta_{55} = 0.255 + 0.040d_1 + 0.099d_2 + 0.201d_3 + 0.392d_4 + 0.353d_5 + 0.144d_6 \\ \eta_{22} = 0.290 + 0.023d_1 + 0.056d_2 + 0.115d_3 + 0.224d_4 + 0.433d_5 + 0.278d_6 \end{cases} \quad (17)$$

It can be concluded from Equations (16b) and (17) that:

1.  The closer the joint is to the beam with the external load applied on, the greater the influencing extent $\lambda_i$ is. Furthermore, after calculation and analysis of $\lambda_i$ of ARHB bridges with different $\gamma$ and different total number of beams, it can be concluded that the influencing extent $\lambda_i$ of joint-*i* on the LLDF of beam-*j* decreases approximately exponentially with an increasing number of beams between joint-*i* and beam-*j*;

2.  When damages occur in joints, for the adjacent beams on both sides of the damaged joint, the LLDF of the beam closer to the external load will be larger;

3.  When the external load $p = 1$ loads onto an interior beam-*k*, for damaged joints with the same distance from beam-*k*, the damaged joint on the side with more beams has the larger impact than the damaged joint on the side with fewer beams on $\eta_{kk}$. For example, for an ARHB bridge with $n = 7$ beams, when the external load $p = 1$ loads onto beam-2 and all joints have the same damage degree, the influencing extent $\lambda_2$ from large to small is $\lambda_2 > \lambda_1 > \lambda_3 > \lambda_4 > \lambda_5 > \lambda_6$. When the external load $p = 1$ is loaded onto beam-3, the influencing extent of each joint is $\lambda_3 > \lambda_2 > \lambda_4 > \lambda_1 > \lambda_5 > \lambda_6$;

4.  For ARHB bridges with the same number of beams, it takes different bending and torsional parameter $\gamma$ values to calculate the LLDFs. Under the same conditions of the external load position, the number of the adjacent rectangular hollow beams, and the damage degree of the joints, the joint with larger $\gamma$ has smaller influencing extent $\lambda_i$ on LLDFs. For example, consider two ARHB bridges A and B with $n = 7$ beams and beam-2 loaded, where the bending and torsional parameter $\gamma$ of bridges A and B are $\gamma_A = 0.1$ and $\gamma_B = 0.05$. Then, the influencing extent $\lambda_{i,A}$ of bridge A is smaller than the influencing extent $\lambda_{i,B}$ of bridge B, i.e., $\lambda_{i,A} < \lambda_{i,B}$. The calculation process will not be repeated.

## 3. Feasibility of the Modified Hinged Connected Beam Method on ARHB Bridges with Diaphragms and Transverse Prestressed Ties

In the United States, Japan, and South Korea, most ARHB bridges are equipped with diaphragms and transverse prestressed reinforcements. For such bridges with reliable transverse connections, accurate LLDFs cannot be obtained by the hinge connected beam method. However, when damage occurs in joints so that the transverse connections of the ARHB bridge become weak, it is feasible to utilize the hinge connected beam method to qualitatively analyze variations in the LLDFs to evaluate the degradation degree of bridge-bearing capacity.

According to *FIB bulletin 43 Structural connections for precast concrete buildings* [28], there are three transmission mechanisms of shear force between adjacent rectangular hollow beams of ARHB bridges: (1) adhesion or friction at joint interfaces, (2) shear-key effect at indented joint faces, and (3) dowel action of transverse steel bars, pins, and bolt crossing joints.

When joints are not cracked, shear force can be transferred by the adhesive bond at joints and the mechanical interlock of shear keys. When cracks arise in joints, the shear force can be transferred by the mechanical interlock of shear keys, dowel action, and the friction enhanced by the pullout resistance of tie bars placed across joints.

In order to explore the contribution of different transmission mechanisms to shear-force transfer, Giraldo-Londoño [29] established a finite-element model with two adjacent rectangular hollow beams connected through a partial-depth shear key. By comparing the differential deflections between adjacent beams of the model with filled transverse post-tensioning ducts and the model with non-filled ones, it can be concluded that dowel action is the main shear force transmission mechanism when the concrete and reinforcements are not yielding, and that shear force transmission performance is not highly affected by variations in the amount of transverse post-tensioning. Zhang [30], Han [31], and Wu [32] have all studied the LLDFs of ARHB bridges before and after external transverse prestressing. It can be concluded that the LLDFs of ARHB bridges with external transverse prestress applied are approximately equal to the LLDFs calculated by reducing the bending and torsional parameter $\gamma$ of the bridge without external transverse prestress. Ge [33] and Cheng [34] analyzed the influence of the installing end and intermediate diaphragms on the LLDFs of fabricated hollow-beam bridges. By comparing the LLDFs of the bridges with and without diaphragms, it can be seen that the LLDFs of the bridge with diaphragms are approximately equal to the ones calculated by reducing $\gamma$ of the bridge without diaphragms.

According to the analysis of the above research data, when no dowel action is provided by transverse prestressed ties, the effect of transverse prestress and diaphragms can be regarded as reducing $\gamma$ to obtain more uniform LLDFs.

For longitudinal prestressed concrete beams, the static flexural rigidity, $EI$, increases with rising post-tensioning force magnitude [35]. Similarly, it is reasonable to deem that the torsional rigidity, $GI_T$, of transversal prestressed adjacent rectangular hollow beams increases with rising transversal post-tensioning force magnitude. Furthermore, the bending and torsional parameter $\gamma$ decreases with an increase in $GI_T$. This assumption is verified by [31–33].

For most ARHB bridges, there is a high risk that the interfaces between shear keys and beams crack due to the load actions from temperature gradient and traffic, and instability workmanship quality. Cracked interfaces slip slightly under the action of shear force, and the resulting small deformation will cause dowel action to counteract the slip and shear force [36]. High concentrated forces are generated in the grouting where the crosswise reinforcements are placed, and considerable tensile stresses may appear in the rest area of the grouting around the crosswise reinforcements [28]. The grouting in the transverse post-tensioning ducts is more likely to fail by the concentrated forces and tensile stresses caused by the dowel action than the concrete in the beam.

Therefore, when cracks and leakage occur at joints, it can be considered that the grouting in the transverse post-tensioning ducts yields at the joint interface. In this case, only when sufficient displacement occurs at the joint can sufficient dowel force be generated to transfer the shear force [36]. However, for small-to-medium-span bridges, the deflection caused by the live load is too small to generate sufficient dowel force [37].

Therefore, for ARHB bridges in service with cracks and leaks at joints, the dowel action caused by transverse prestressed reinforcements can be considered to have failed, and the lateral force transmission mechanism of the bridge will be similar to that of ARHB bridges without diaphragms and transverse prestressed ties. The effects of the joint damage degree on LLDFs can be analyzed by a modified hinge connected beam method to assess the degradation degree of ARHB bridge-bearing capacity.

## 4. Lateral Load Distribution Performance Rating Number *LDN*

Damage on grouted joints and beams lead to the degradation of lateral load transmission performance, which can cause variations in the LLDFs of the bridges. In the service stage of the ARHB bridges, the grouted joints are always damaged before the beams. The

degradation of lateral load transmission performance will increase the failure probability of adjacent rectangular hollow beams. According to "the law of fives", when damage occurs in the bridge, the earlier the repairs and maintenances are carried out, the lower the cost will be. Timely repairs and maintenance of damaged grouted joints can effectively avoid the degradation of beams. As joint damage is common in ARHB bridges, a low-cost assessment method that can be combined with conventional visual inspection is needed to evaluate the bearing performance of bridges and guide maintenance schedules.

This paper proposes the lateral load distribution performance rating number $LDN$, which is based on the concept of standard deviation, to assess the degradation degree of bearing capacity of ARHB bridges with damaged joints and generally intact beams, and is shown as Equation (18):

$$
\begin{cases}
LDN = S \times \sqrt{\dfrac{\left(\sum_{i=1}^{i=N} VA_i\right)^2}{N}} \times 100 \\[3mm]
LDN_{pre} = S \times \sqrt{\dfrac{\left(\sum_{i=1}^{i=N} VA_{i,pre}\right)^2}{N}} \times 100
\end{cases}
\tag{18}
$$

where $VA_i$ represents the variation in the LLDF of the beam being analyzed, i.e., beam-$i$, in the bridge without transverse interior prestressed reinforcement; $VA_{i,pre}$ represents the one of beam-$i$ in the bridge with transverse interior prestressed reinforcement; $S$ is the coefficient representing the design safety grade of the bridge and is determined by the *General Specifications for Design of Highway Bridges and Culverts (JTGD60-2015)* [22]; see Table 1 for values. $N$ represents the total number of adjacent rectangular hollow beams of the bridge.

**Table 1.** $S$ values according to design safety grade.

| Design Safety Grade | $S$ | Applicable Objects |
| --- | --- | --- |
| 1 | 1.1 | (1) Super large bridges, major bridges, and medium bridges on highways of all grades; (2) Small bridges on freeways, Grade I highways, Grade II highways, national defense highways, and busy roads near cities. |
| 2 | 1 | (1) Small bridges on Grade III and IV highways; (2) Culverts on expressways, Grade I highways, Grade II highways, national defense highways, and highways with heavy traffic near cities. |
| 3 | 1 | Culverts on Grade III and IV highways. |

The calculation formulas of $VA_i$ and $VA_{i,pre}$ are shown as Equation (19a,b):

$$
VA_i = \frac{\sum_{n=1}^{n=N_\gamma} (S_n \times D_n \times P_n)}{\sum_{n=1}^{n=N_\gamma} \left(S_{n,\,ref} \times D_{nL,ref} \times P_n\right)}
\tag{19a}
$$

$$
VA_{i,pre} = K_1 \frac{\sum_{n=1}^{n=N_\gamma} (S_n \times D_n \times P_n)}{\sum_{n=1}^{n=N_\gamma} \left(S_{n,\,ref} \times D_{nL,ref} \times P_n\right)} + K_2
\tag{19b}
$$

where $K_1 = 0.7$ is the weight coefficient of joints and $K_2 = 0.3$ is the weight coefficient of dowel action. The determinaiton of values of $K_1$ and $K_2$ is based on [29].

$D_n$ represents the damage degree of joint-$n$, and the value is presented in Table 2; $D_{n,ref}$ is the corresponding maximum value.

**Table 2.** $D_n$ values representing the damage degree of joint-$n$.

| The Degree of Damage | $D_n$ |
| --- | --- |
| Intact | 0 |
| Slight | 1 |
| Medium | 2 |
| Severe | 3 |

$N_\gamma$ is the number of the joints whose influencing extent on beam-*i* is considered, and the value is determined by the bending and torsional parameter $\gamma$; when $\gamma \leq 0.25$, $N_\gamma = 5$, and when $\gamma > 0.25$, $N_\gamma = 3$.

$S_n$ is the influencing extent coefficient of joint-*n* on the LLDF of beam-*i*; its value depends on $N_\gamma$ and the number of beams spaced between joint-*n* and beam-*i*; the value of $S_n$ is presented in Table 3; $S_{n,ref}$ is the corresponding maximum value.

**Table 3.** $S_n$ values representing the influencing extent of joint-n on the LLDF of beam-*i*.

| The Number of Beams Spaced between the Joint and the Beam Being Analyzed | $S_n$ ($N_\gamma$=5) | $S_n$ ($N_\gamma$=3) |
|:---:|:---:|:---:|
| 0 | 5 | 4 |
| 1 | 3 | 2 |
| 2 | 2 | 1 |
| 3 | 1.5 | / |
| 4 | 1 | / |

$P_n$ is the position coefficient of joint-*n*; if the damaged joints occur in pairs on both sides, $P_n = 0.6$ for joints located on the side with more beams, $P_n = 0.4$ for joints located on the side with fewer beams, and $P_n = 0.5$ where the numbers of beams on both sides are the same; when the beam being analyzed is the exterior beam or there is no corresponding joint on the other side, $P_n = 1$. For example, when studying beam-3 of an ARHB bridge with nine beams, if the numbers of damaged joints are joint-1 to 3 and 5 to 7, the position coefficients are shown in Table 4.

**Table 4.** $P_n$ values of an ARHB bridge with 9 beams.

| Number of Joints | Joints Status | $P_n$ |
|:---:|:---:|:---:|
| 1 | damaged | 0.4 |
| 2 | damaged | 1 |
| 3 | intact | - |
| 4 | damaged | 0.6 |
| 5 | damaged | 1 |
| 6 | damaged | 1 |
| 7 | damaged | 1 |
| 8 | - | - |

Note: when studying beam-3, the influence of joint-8 is not considered, so the data of joint-8 are not shown in Table 3.

The value of $D_n$, $N_\gamma$, $S_n$, and $P_n$ is determined by the expert evaluation method.

It should be noted that the method proposed in this article only evaluates the influence of transverse connections, namely, grouted joints and transverse prestressed reinforcement, on the lateral load transmission performance of bridge superstructure. This does not consider the lateral load redistribution caused by the performance degradation of the adjacent rectangular hollow beam. In addition, this method does not involve the performance assessment of the substructure and deck system of the bridge.

## 5. Case Study

### 5.1. Introduction of the Bridges

#### 5.1.1. Changhong Tunnel K1 + 812 Bridge on East Outer Ring Road

Changhong Tunnel K1 + 812 Bridge is a two-span simply supported ARHB bridge of 10 m + 13 m, composed of 18 adjacent rectangular hollow beams. The superstructure of the 13 m span comprises prestressed concrete beams with 62 cm height and 40 MPa compressive strength, and the superstructure of the 10 m span is reinforced concrete beams with 52 cm height and 30 MPa compressive strength. The width of interior beams is 99 cm and of exterior beams is 99.5 cm. The length of flanges of exterior beams is 50 cm. The

bridge deck is paved in two layers. The lower layer is paved with 7 cm-thick 30 MPa reinforced concrete, and the upper layer is paved with 6 cm-thick asphalt concrete.

The 13 m span is located in the south and the 10 m in the north. The beams and grouted joints are numbered from east to west. For example, the beam and the grouted joint at the easternmost side of the 13 m span are beam-S1 and joint-S1, and the beam and the grouted joint at the westernmost side of the 10 m span are beam-N18 and joint-N17. This is shown in Figure 3a.

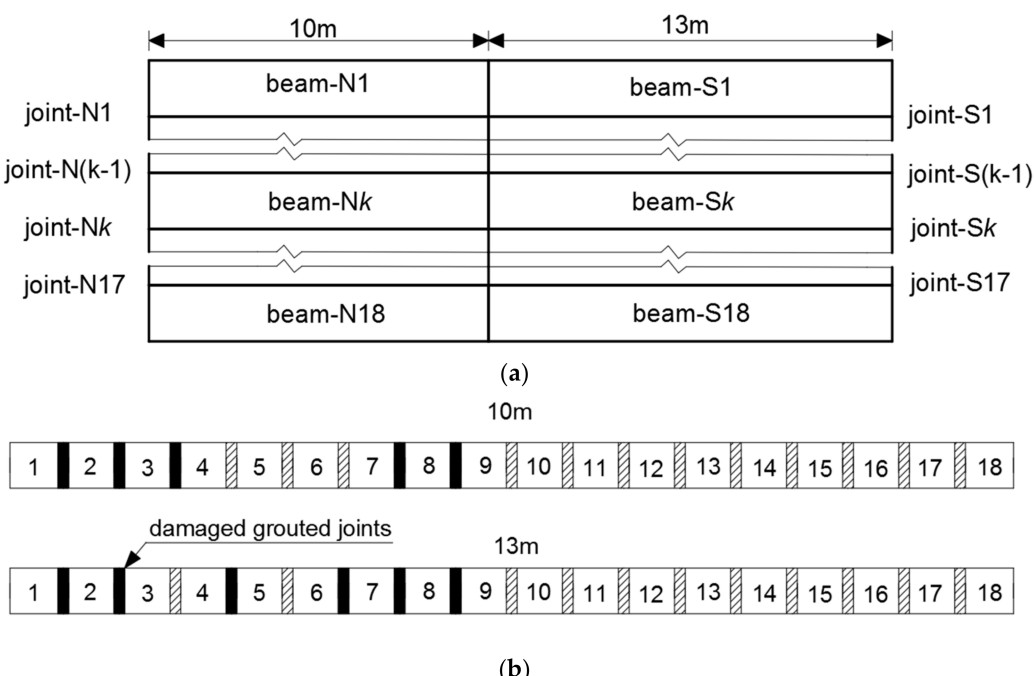

**Figure 3.** (**a**) Arrangement of numbers of beams and joints of Changhong Tunnel K1 + 812 Bridge. (**b**) Location of damaged joints of Changhong Tunnel K1 + 812 Bridge.

The inspection of the bridge shows that there is exfoliation of the concrete and corrosion of the reinforcing steel bars at the bottom of the flange of beam-S18; leakage exists in joint-S1, S2, S4, S6, S7, and S8, and in joint-N1, N2, N3, N7, and N8. There are three longitudinal cracks in each of the two span bridges, all located near the grouted joints, as shown in Figure 3b.

According to the inspection report produced in 2012 based on the *Technical code of maintenance for city bridge (CJJ 99-2003)* [38], the assessment grades of the superstructures of the two spans of the bridge are both D.

5.1.2. Baidong No. 2 Bridge on Baizhang East Road

Baidong No. 2 Bridge is a single-span simply supported ARHB bridge with a span of 13 m. The carriageway is composed of 17 adjacent rectangular hollow concrete beams with 65 cm height, 99 cm width, and 25 MPa compressive strength, and the sidewalks on both sides are π-beams. The bridge deck is paved in two layers. The lower layer is paved with 6 cm-thick concrete, and the upper layer is paved with 2 cm-thick asphalt concrete.

The beams and grouted joints are numbered from south to north as beam-1 to beam-17 and joint-1 to joint-16, as shown in Figure 4a.

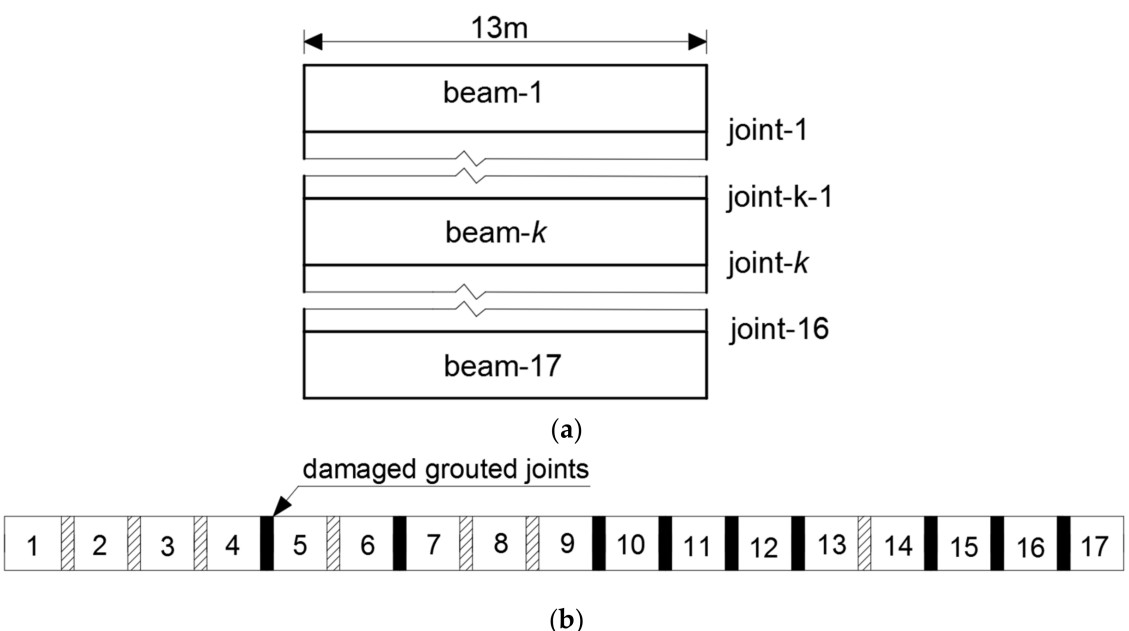

**Figure 4.** (**a**) Arrangement of numbers of beams and joints of Baidong No. 2 Bridge. (**b**) Location of damaged joints of Baidong No. 2 Bridge.

The inspection of the bridge shows that there is exfoliation of the concrete and reinforcement corrosion on the side of beam-1, and leakage exists in joint-4, 6, 9, 10, 11, 12, 14, 15, and 16, as shown in Figure 4b.

According to the inspection report produced in 2012 based on the *Technical code of maintenance for city bridge (CJJ 99-2003)* [38], the assessment grade of the superstructure of the bridge is D.

5.1.3. Yunhe Bridge on Huancheng North Road

Yunhe Bridge is a five-span simply supported ARHB bridge of 11 m + 4 m × 20 m + 4 m × 25 m. The right half of the bridge is composed of 18 beams, of which the carriageway is composed of 14 adjacent rectangular hollow beams with 60 cm height, 99 cm width, and 25 MPa compressive strength, and the sidewalks on both sides are composed of two hollow beams and sidewalk slabs. The upper structure of the 11 m span is reinforced concrete beams and the others are prestressed concrete beams.

The beams are numbered from south to north as beam-1 to beam-18, and beam-3 to beam-16 are the numbers of the beams that make up the carriageway. The grouted joints are numbered from joint-1 to joint-13, as shown in Figure 5a.

The inspection of the bridge shows that, at the span of 11 m, there is exfoliation of the concrete and reinforcement corrosion on the side of beam-3, all joints are partially detached, and there is no leakage, as shown in Figure 5b.

According to the inspection report produced in 2013 based on the *Technical code of maintenance for city bridge (CJJ 99-2003)* [38], the assessment grade of the superstructure of the 11 m span of the bridge is D.

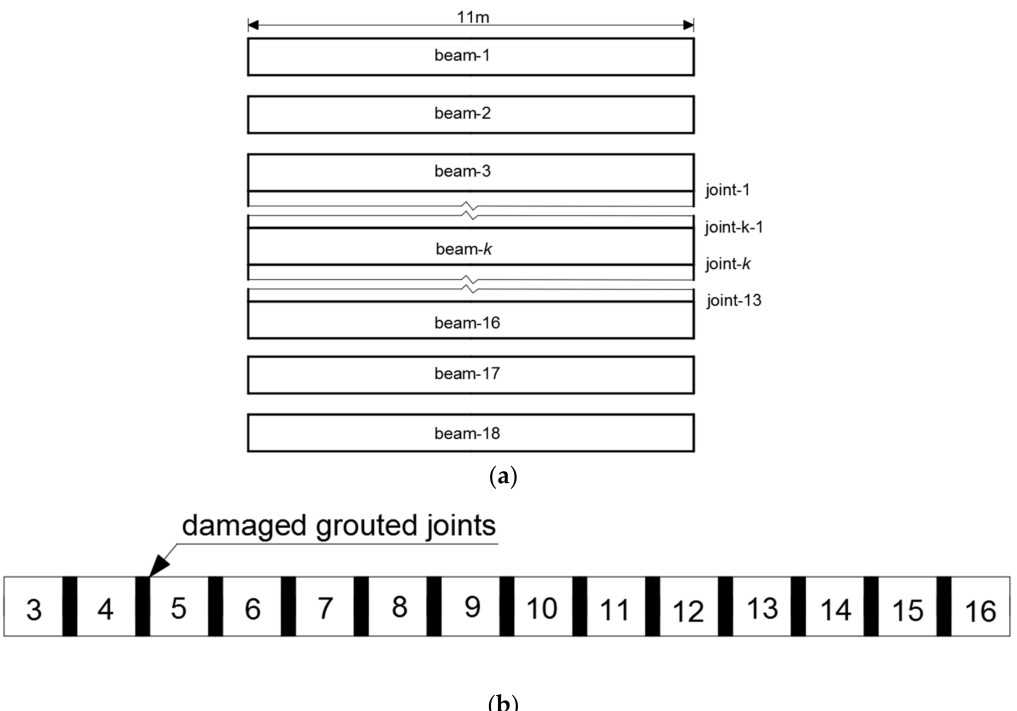

**Figure 5.** (**a**) Arrangement of numbers of beams and joints of Yunhe Bridge. (**b**) Location of damaged joints of Yunhe Bridge.

*5.2. Lateral Load Distribution Performance Rating Number LDN of the Three Bridges*

It can be seen from the introduction in Section 5.1 that the assessment grades of the superstructure of the three bridges are all D. All three bridges show exfoliation of concrete and reinforcement corrosion. The grouted joints of each bridge are damaged to various degrees.

In order to establish an appropriate maintenance sequence, this paper uses the method proposed in Section 3 to calculate the lateral load distribution performance rating number *LDN* of the three bridges.

The importance coefficient of the three bridges is $S = 1.1$. The bending and torsional parameter of the three bridges is $\gamma \leq 0.25$, so $N_\gamma = 5$. There are no transverse ties for the three bridges, so Equation (18a) is selected for the calculation.

According to the visual inspection results:

1. For the 13 m span of Changhong Tunnel K1 + 812 Bridge, the damage degree $D_n$ of the damaged joints is $D_n = 3$, where $n = 1, 2, 4, 6, 7, 8$, and the other joints are intact with $D_n = 0$.; for the 10 m span, $D_n$ of the damaged joints is $D_n = 3$, where $n = 1, 2, 3, 7, 8$, and the other joints are intact with $D_n = 0$;
2. For Baidong No. 2 Bridge, the damage degree $D_n$ of the damaged joints is $D_n = 3$, where $n = 4, 6, 9, 10, 11, 12, 14, 15$, and the others are intact with $D_n = 0$;
3. For Yunhe Bridge, the damage degree $D_n$ of all joints is $D_n = 1$, where $n = 1$ to 13.

Before calculating the lateral load distribution performance rating number *LDN* of the three bridges, Equation (19a) was used to calculate the variation in the LLDF $VA_i$ of each beam of the three bridges. The results are shown in Table 5a–c. For Changhong Tunnel K1 + 812 Bridge, due to the lack of truck testing data that could be compared, only $VA_i$ of the 13 m span is displayed in Table 5a.

**Table 5.** (**a**) $VA_i$ of the 13 m span of Changhong Tunnel K1 + 812 Bridge. (**b**) $VA_i$ of Baidong No. 2 Bridge. (**c**) $VA_i$ of Yunhe Bridge.

| (a) | | | |
|---|---|---|---|
| **Beam-*n*** | $VA_i$ | **Beam-*n*** | $VA_i$ |
| Beam-1 | 0.761 | Beam-10 | 0.312 |
| Beam-2 | 0.555 | Beam-11 | 0.216 |
| Beam-3 | 0.600 | Beam-12 | 0.120 |
| Beam-4 | 0.617 | Beam-13 | 0.048 |
| Beam-5 | 0.584 | Beam-14 | 0 |
| Beam-6 | 0.656 | Beam-15 | 0 |
| Beam-7 | 0.641 | Beam-16 | 0 |
| Beam-8 | 0.544 | Beam-17 | 0 |
| Beam-9 | 0.352 | Beam-18 | 0 |
| **(b)** | | | |
| **Beam-*n*** | $VA_i$ | **Beam-*n*** | $VA_i$ |
| Beam-1 | 0.120 | Beam-10 | 0.664 |
| Beam-2 | 0.240 | Beam-11 | 0.768 |
| Beam-3 | 0.264 | Beam-12 | 0.784 |
| Beam-4 | 0.336 | Beam-13 | 0.760 |
| Beam-5 | 0.352 | Beam-14 | 0.760 |
| Beam-6 | 0.456 | Beam-15 | 0.856 |
| Beam-7 | 0.440 | Beam-16 | 0.840 |
| Beam-8 | 0.504 | Beam-17 | 0.880 |
| Beam-9 | 0.580 | - | - |
| **(c)** | | | |
| **Beam-*n*** | | $VA_i$ | |
| All beams | | 0.333 | |

It can be seen from Table 6 that the lateral load distribution performance rating number *LDN* of Baidong No. 2 Bridge is the largest among the three bridges, which means that the variation in the LLDFs of the beams is the largest, and that the degradation of the load transmission performance of the bridge is the worst. Next is Changhong Tunnel K1 + 812 Bridge. Yunhe Bridge has the least degradation in the load transmission performance.

**Table 6.** *LDN* of Changhong Tunnel K1 + 812 Bridge, Baidong No. 2 Bridge, and Yunhe Bridge.

| **Bridge** | **LDN** |
|---|---|
| Changhong Tunnel K1 + 812 Bridge | $LDN_1 = 43.40$ |
| Baidong No. 2 Bridge | $LDN_2 = 67.41$ |
| Yunhe Bridge | $LDN_3 = 36.66$ |

Therefore, the sequencing in bridge maintenance from first to last should be Baidong No. 2 Bridge, Changhong Tunnel K1 + 812 Bridge, and Yunhe Bridge.

*5.3. Comparison between Calculation Results and Static Load Test Results*

In this section, the results obtained in Section 5.2 are verified by the static load test results of the three bridges.

The static load test results of inspection reports of Baidong No. 2 Bridge and Changhong Tunnel K1 + 812 Bridge give the measured and theoretical values of the LLDF of each beam under central load conditions and two kinds of eccentric load conditions. The inspection report of Yunhe Bridge gives the theoretical and measured values of mid-pan deflection of each beam under central load conditions and two kinds of eccentric load conditions,

which can be used to calculate the LLDF of each beam. See Figure 6a–c for lateral deflection influence lines obtained for the three tested bridges.

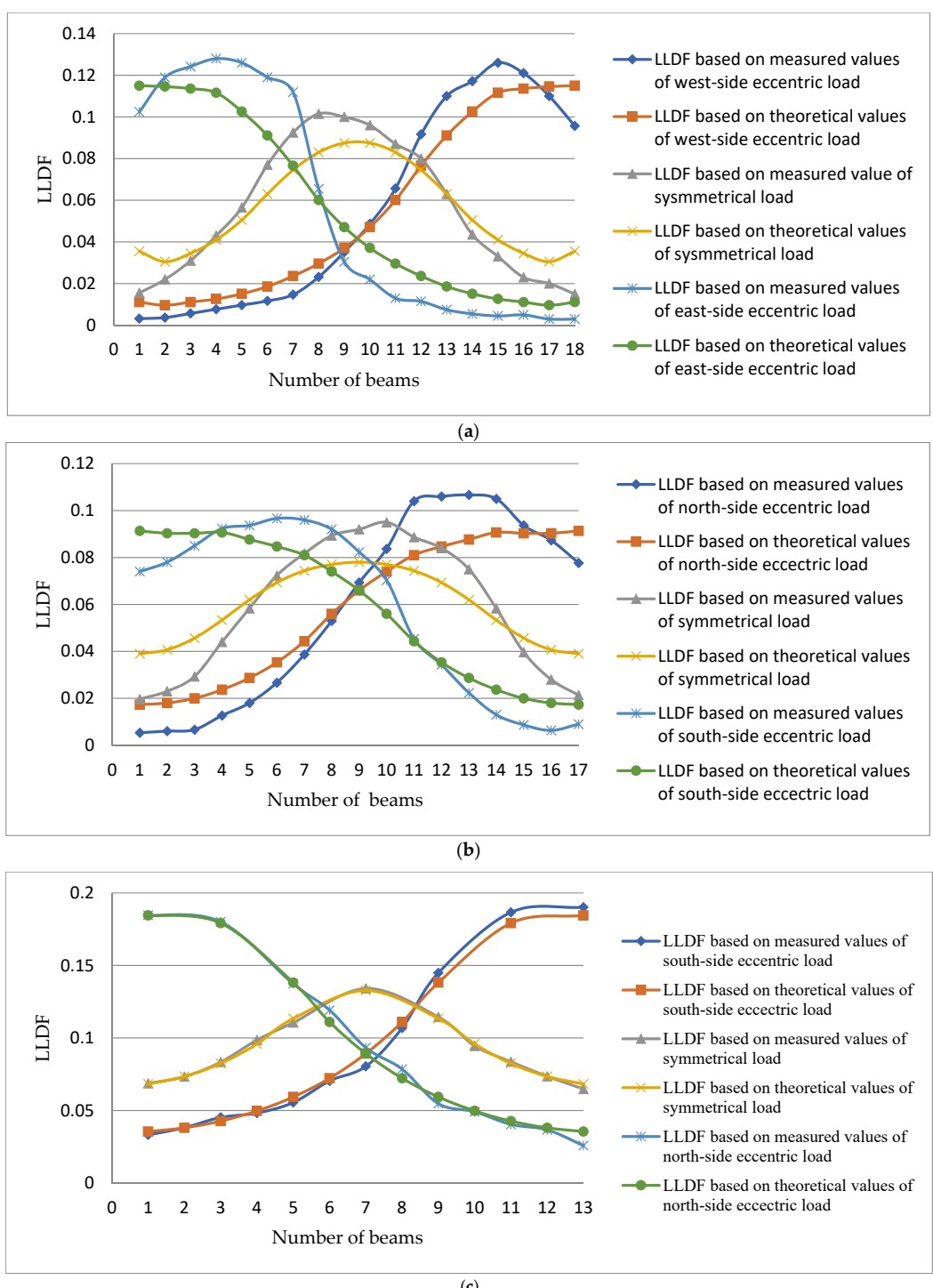

**Figure 6.** (**a**) Lateral deflection influence lines of Changhong Tunnel K1 + 812 Bridge (13 m span). (**b**) Lateral deflection influence lines of Baidong No. 2 Bridge. (**c**) Lateral deflection influence lines of Yunhe Bridge.

Equation (19) is proposed based on the concept of standard deviation to calculate the variation degree of the LLDFs of the three bridges. The result is shown in Table 7.

$$VD_{ldf} = \sqrt{\frac{\left(\sum_{i=1}^{i=N} V_{ldf,i}\right)^2}{N}} \tag{19}$$

where $VD_{ldf}$ represents the variation degree of the LLDFs of the bridge, $V_{ldf,i}$ represents variation in the LLDF of beam-$i$, and $N$ is the total number of beams. The selection principle of $V_{ldf,i}$ is that for each beam, choose the largest difference among the three load conditions.

**Table 7.** $VD_{ldf}$ of Changhong Tunnel K1 + 812 Bridge, Baidong No. 2 Bridge, and Yunhe Bridge.

| Bridge | $VD_{ldf}$ |
|:---:|:---:|
| Changhong Tunnel K1 + 812 Bridge | $VD_{ldf,1} = 0.012$ |
| Baidong No. 2 Bridge | $VD_{ldf,2} = 0.016$ |
| Yunhe Bridge | $VD_{ldf,3} = 0.0055$ |

The sequencing in bridge maintenance determined by Table 7 is the same as the result obtained in Section 5.2.

## 6. Conclusions

In this paper, a joint damage-based assessment indicator for assessing the degradation degree of lateral load transmission performance of ARHB bridges is proposed.

This paper modifies the traditional hinge connected beam method by introducing relative displacement in grouted joints. The relationship between joint position and LLDFs is analyzed by a modified hinge connected beam method, and summarized as that the influencing extent of joint-$i$ on beam-$j$ decreases approximately exponentially with an increasing number of beams between joint-$i$ and beam-$j$.

This paper proposes a low-cost assessment method that can be combined with visual inspection. This method uses a new assessment indicator, namely, lateral load transmission performance rating number $LDN$, to assess the degradation degree of load-bearing capacity of ARHB bridge superstructure. The assessment result can be used to determine the maintenance sequence. The accuracy and feasibility of the proposed method have been verified by static load test results of three concrete ARHB bridges.

Further developments are expected considering the influence of material deterioration of beams on LLDFs of ARHB bridges in order to be more suitable for the needs of actual projects.

**Author Contributions:** Methodology, S.W.; validation, J.D.; investigation, S.W.; resources, J.D.; data curation, J.D.; writing—original draft preparation, S.W.; writing—review and editing, H.S.; supervision, J.D. All authors have read and agreed to the published version of the manuscript.

**Funding:** This research was funded by the Open Project of National Engineering Laboratory of Bridge Structure Safety Technology (Beijing), grant number 2020-GJKFKT-6, Transportation Science and Technology Project of Shaanxi Transportation Department of China, grant number 20-10K, and the National Natural Science Foundation of China—National Project for Research and Development of Major Scientific Instruments, grant number 51727813.

**Institutional Review Board Statement:** Not applicable.

**Informed Consent Statement:** Not applicable.

**Data Availability Statement:** Not applicable.

**Conflicts of Interest:** The authors declare no conflict of interest. The funders had no role in the design of the study; in the collection, analyses, or interpretation of data; in the writing of the manuscript; or in the decision to publish the results.

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
