# Peer review of "Low-Cost Assessment Method for Existing Adjacent Beam Bridges"

_applsci, doi:10.3390/app122111304_

Round 1
Reviewer 1 Report
The Introduction and mainly the part 2 is too large. I do not know if all equations should be mentioned in text.
In Table 5-1A same values are not given with 3 numbers after decimal poit - as the majority of values are in this style, it would be better used the same style althouth they were zero number - 0,120 better than 0,12 only.
in text on page 17 the Table 6 is mentioned. There is no Table 6 in all paper - please correct numebring or include Table 6.
Author Response
Point 1: The Introduction and mainly the part 2 is too large. I do not know if all equations should be mentioned in text.
Thank you for the reviewing and the commendation, the comments are responded in the following, the manuscript has been revised accordingly, and all changes are highlighted in yellow.
Response 1: The introduction and the part 2 have been simplified, and the deleted parts have been marked with strikethrough and highlighted in yellow.
Point 2: In Table 5-1A same values are not given with 3 numbers after decimal poit - as the majority of values are in this style, it would be better used the same style althouth they were zero number - 0,120 better than 0,12 only.
Response 2: The values in Table 5-1A have been modified.
Point 3: in text on page 17 the Table 6 is mentioned. There is no Table 6 in all paper - please correct numebring or include Table 6.
Response 3: Table 6 has been added.

Reviewer 2 Report
Dear Editor:
The followings are my comments regarding the manuscript entitled “
"Low Cost Assessment Method for Existing Adjacent Beam Bridges"
Abstract: give more detail on accuracy of the calculation, number of sampling, parameters taking into account, percentage increase of cost
1- Line 20: give detail on the “new low cost assessment method and a new assessment index”. Required more detail
2- Line 22: give detail on how you verification your new low cost assessment method
3. Feasibility of Modified Hinged Connected Beam Method on ARHB Bridges with 461 Diaphragms and Transverse Restressed Ties
1- Line496: “analysis of the above research data” give more detail on the data limitation and indicate the references.
2- Line 499: What you mean by flexural rigidity, ?I? Give more detail on the percentage of the increases.
3- Line 501: What you mean by torsional rigidity, ??T? Give more detail on the percentage of the increases.
4. Lateral Load Distribution Performance Rating Number ???
1- Line 539: Why do you use equation (18) give code requirement or any protocol for that.
2- Line540 : In general give the limitation of the equation No. 18
3- Line560: why you use ?1 = 0.7; ?2 = 0.3 as the weight coefficient of dowel action in equation (19-1) and (19-2).
4- Line577: In Table 2 give more information and limitation on “The degree of the damage”
Case Study
In general, give more detail on. Longitudinal (top) and cross sections (bottom) of the concrete Bridges in this study and highlight the exfoliation of the concrete, damage beam, and reinforcement corrosion on the side of beam.
Author Response
Point 1:
Abstract
Line 20: give detail on the “new low cost assessment method and a new assessment index”. Required more detail
Response1:
Thank you for the reviewing and the commendation, the comments are responded in the following, the manuscript has been revised accordingly, and all changes are highlighted in yellow.
More details have been added.
Point 2:
Line 22: give detail on how you verification your new low cost assessment method
Response2: More details have been added.
Point 3:
- Feasibility of Modified Hinged Connected Beam Method on ARHB Bridges with 461 Diaphragms and Transverse Restressed Ties
Line496: “analysis of the above research data” give more detail on the data limitation and indicate the references.
Response3: References have been indicated in the corresponding position of the manuscript. Because this paper is not an accurate analysis or calculation of the degradation degree of lateral bearing capacity of ARHB bridges, it only needs to analyse the influencing trend of the effect of transverse prestress and diaphragms on LLDF, and does not need to calculate the specific data limitation.
Point 4:
Line 499: What you mean by flexural rigidity, ?I? Give more detail on the percentage of the increases.
Response4: Flexural rigidity EI refers to the flexural rigidity of the cross section of the beam perpendicular to the longitudinal axis. In this paper, the influence of post-tensioning force magnitude of longitudinal prestressed reinforcements on flexural rigidity is used to derive the influence of transverse prestress ed reinforcements on torsional rigidity, which is a qualitative analysis. so it is not necessary to determine specific values. More details can be found in [1].
Point 5:
Line 501: What you mean by torsional rigidity, ??T? Give more detail on the percentage of the increases.
Response5: Torsional rigidity GIT refers to torsional rigidity of the cross section of the beam perpendicular to the longitudinal axis. The exact change value of is related to the value of prestressing force, the specific design parameters of the beams and the position of the prestressed ties. This paper is a qualitative analysis, and does not need to determine the exact value of GIT.
Point 6:
Lateral Load Distribution Performance Rating Number ???
Line 539: Why do you use equation (18) give code requirement or any protocol for that.
Response6: Equation (18), which has been revised to Equation (17), is based on the concept of standard deviation and is added in corresponding position.
Point 7:
Line540 : In general give the limitation of the equation No. 18
Response7: Equation (18) , which has been revised to Equation (17) is only applicable to the assessment of the degradation degree of the lateral bearing capacity of AHRB bridges. The lateral load distribution performance rating number proposed in this paper can not reflect the exact value of lateral bearing capacity degradation, but can be used for comparision among the degradation degree of bridges.
Point 8:
Line560: why you use ?1 = 0.7; ?2 = 0.3 as the weight coefficient of dowel action in equation (19-1) and (19-2).
Response8: The determinaiton of values of and is based on [2].
Point 9:
Line577: In Table 2 give more information and limitation on “The degree of the damage”
Response9: Values in Table 2 are determined by expert evaluation method.
Point 10:
Case Study
In general, give more detail on. Longitudinal (top) and cross sections (bottom) of the concrete Bridges in this study and highlight the exfoliation of the concrete, damage beam, and reinforcement corrosion on the side of beam.
Response10:
The details of longitudinal (top) and cross sections (bottom) of the bridges have been added.
The beam number with exfoliation and reinforcement corrosion has been described in the introduction of the bridges, but due to the lack of data in the bridge inspection report, the exact coordinates of the damages cannot be obtained. In addition, the exact coordinates of exfoliation and reinforcement corrosion do not affect the calculation of LDN.
Reference:
- Noble D, Nogal M, Pakrashi V. The effect of prestress force magnitude and eccentricity on the natural bending frequencies of uncracked prestressed concrete beams[J]. Journal of Sound and Vibration, 2016, 365: 22-44.
- Giraldo-Londoño O. Finite element modeling of the load transfer mechanism in adjacent prestressed concrete box-beams[D]. Ohio University, 2014.

Reviewer 3 Report
The article is valuable because it presents a cost-effective method of assessing used and partially damaged ARHB (adjacent rectangular hollow beam) bridges for their further operation.
I have some comments:
1. What method was used for determining the coefficients presented in Tables 1, 2, 3 ? I suppose it was not trial and error. Please note that I do not expect all technical details of the method used.
2. Table 6 is missing. Please add.
3. Lateral deflection influence lines obtained for three tested bridges (described in Section 4) would be a good graphic illustration of the case studies.
4. I feel that three case studies are far too few to accept the method as correct (see lines from 722 to 724).
5. Title of subsection 2.1 is missing. Please add.
Author Response
Point 1:
What method was used for determining the coefficients presented in Tables 1, 2, 3 ? I suppose it was not trial and error. Please note that I do not expect all technical details of the method used.
Response 1:
Thank you for the reviewing and the commendation, the comments are responded in the following, the manuscript has been revised accordingly, and all changes are highlighted in yellow.
The coefficients presented in Tables 1 are determined by General Specifications for Design of Highway Bridges and Culverts(JTGD60-2015), and explained in the corresponding position of the manuscript. The coefficients presented in Tables 2 and Table 3 are determined by expert evaluation method and explained in the corresponding position of the manuscript.
Point 2:
Table 6 is missing. Please add.
Response 2:
Table 6 has been added.
Point 3:
Lateral deflection influence lines obtained for three tested bridges (described in Section 4) would be a good graphic illustration of the case studies.
Response 3:
Lateral deflection influence lines obtained for three tested bridges have been added to the manuscript.
Point 4:
I feel that three case studies are far too few to accept the method as correct (see lines from 722 to 724).
Response 4:
The degradation degree of LLFD is affected both by joint damages and beams damages. When selecting case studies to verify the propsed method, it is necessary to ensure that the damage degree of beams of the bridges is not much different, so that the impact extent of joint damage degree on LLDF can be reflected correctly. At present, in the special bridge inspection reports (with static load test that can be used to virified the proposed method) available to the author, most bridges have serious girder damage. In the future research, further data collection will be conducted by the author to supplement.
Point 5:
Title of subsection 2.1 is missing. Please add.
Response 5:
Title of subsection 2.1 has been add.
